# Is There a New Road to Spinal Cord Injury Rehabilitation? A Case Report about the Effects of Driving a Go-Kart on Muscle Spasticity

**DOI:** 10.3390/diseases11030107

**Published:** 2023-08-22

**Authors:** Giacomo Farì, Maurizio Ranieri, Riccardo Marvulli, Laura Dell’Anna, Annatonia Fai, Lucrezia Tognolo, Andrea Bernetti, Laura Caforio, Marisa Megna, Ernesto Losavio

**Affiliations:** 1Department of Translational Biomedicine and Neuroscience (DiBraiN), Aldo Moro University, 70121 Bari, Italy; maurizio.ranieri@uniba.it (M.R.); riccardo.marvulli@policlinico.ba.it (R.M.); lauradellanna@gmail.com (L.D.); annat.fai@gmail.com (A.F.); laura.caforio@uniba.it (L.C.); marisa.megna@uniba.it (M.M.); 2Department of Biological and Environmental Science and Technologies (Di.S.Te.B.A.), University of Salento, 73100 Lecce, Italy; 3Rehabilitation Unit, Department of Neuroscience, University of Padova, 35100 Padova, Italy; lucrezia.tognolo@unipd.it; 4Department of Anatomy, Histology, Forensic Medicine and Orthopedics, Sapienza University, Piazzale Aldo Moro 5, 00185 Rome, Italy; andrea.bernetti@uniroma1.it; 5Neurorehabilitation and Spinal Unit, Clinical and Scientific Institutes Maugeri IRCCS, 70124 Bari, Italy; ernesto.losavio@icsmaugeri.it

**Keywords:** musculoskeletal, spasticity, whole-body vibration, adapted sports, endocrine system, therapeutic strategies

## Abstract

Background: Traumatic spinal cord injury (SCI) is a neurological disorder that causes a traumatic anatomical discontinuity of the spinal cord. SCI can lead to paraplegia, spastic, or motor impairments. Go-karting for people with SCI is an adapted sport that is becoming increasingly popular. The purpose of this case report is to shed light on the effects of driving a go-kart on a patient with SCI-related spasticity and to deepen understanding of the possible related role of whole-body vibration (WBV) and neuroendocrine reaction. Methods: The patient was a 50-year-old male with a spastic paraplegia due to traumatic SCI. He regularly practiced go-kart racing, reporting a transient reduction in spasticity. He was evaluated before (T0), immediately after (T1), 2 weeks after (T2), and 4 weeks after (T3) a go-kart driving session. On both sides, long adductor, femoral bicep, and medial and lateral gastrocnemius spasticity was assessed using the Modified Ashworth Scale (MAS), and tone and stiffness were assessed using MyotonPro. Results: It was observed that a go-kart driving session could reduce muscle spasticity, tone, and stiffness. Conclusions: Go-kart driving can be a valid tool to obtain results similar to those of WBV and hormone production in the reduction of spasticity.

## 1. Introduction

Traumatic spinal cord injury (SCI) is a neurological life-changing condition [1], which can lead to paraplegia or quadriplegia [2]. It causes a traumatic anatomical discontinuity in the spinal cord, resulting in interruption of impulse conduction and in permanent damage to the spinal cord’s normal functions [3]. The incidence of SCI is higher in men compared to women and has a peak of incidence in young adults, with severe psychological impacts since an affected person will have to adapt their life to disability [4,5]. Moreover, for people with SCI, mortality is two to five times higher than that of the general population, with worse survival rates in poor countries [6,7].

Spasticity is a very frequent consequence of SCI [8]; it is estimated that 67–78% of individuals with SCI experience spasticity after the acute phase [9]. In particular, spastic hypertonia is associated with increased reflex excitability and disorder of motor output, that involves abnormal increase in muscle tone and velocity-dependent increased resistance to muscle passive stretch. The loss of presynaptic inhibition caused by interruption of the spinal cord is indicated by increased amplitude of the Hoffman reflex (H-reflex) in individuals with spasticity [10]. Clonus, the clasp-knife phenomenon, ‘Babinski’ signs, muscles spasms, exaggerated reflexes, autonomic dysfunctions, and dystonia are some of the clinical consequences of spastic hypertonia [11].

The most used and least invasive way to treat spasticity is through rehabilitation. This is often considered to be complementary to pharmacological therapies, that may be associated with negative effects, such as muscle weakness and drowsiness, which can further impair motor function, and to surgical therapies [12,13]. In recent years, physical therapeutic treatments have gained increasing attention as neuromodulatory approaches and non-pharmacological alternatives for alleviating spasticity [14].

Rehabilitative interventions used in spasticity management are various; they include treatments focused on stretching exercises to preserve muscle length [15], therapeutic exercises to improve muscle strength [16], ultrasound therapy [17] and extracorporeal shock waves, which seem useful for reducing pain and improving neuromuscular functions [18], and electrical neuromuscular stimulation to induce synaptic plasticity and to promote the production of endogenous neurotrophic factors [19]. Also, the application of deep learning can assist spasticity rehabilitation in estimating movement variables (i.e., muscle forces and joint moments) that cannot be easily measured in vivo and can enable better interpretation of the dynamic interaction between neural impulses and muscles [20,21].

Moreover, both focal and whole-body vibration (WBV) represent therapeutic treatments for spasticity, activating presynaptic inhibitory inputs that reduce the excitatory influence of Ia afferent inputs [22].

There are, therefore, current challenges for spasticity management in rehabilitation; discovering new instruments that can assist this is crucial for better functional recovery of these patients, with the aim of reducing the impact of their motor disability.

Sports practice offers many advantages for people suffering from an SCI, such as improvement in social participation and the quality of life (QoL) [23] and better acceptance of physical disability [24]. The main question is: can adapted sport reduce spasticity? Several studies have affirmed the positive impact of sports like yoga, Pilates [25], wheelchair sports [26], and even climbing [27], on spasticity related to different neurological diseases, but there is still a lack of scientific evidence concerning the neurophysiological mechanisms and duration of muscle relaxant effects.

Among adapted sports, go-karting for people with SCI is now becoming more and more popular. It is practiced on tracks using small four-wheeled adapted vehicles. During the drive, because of the intrinsic engine and vehicle characteristics, an intense and non-measurable dose of WBV is transmitted to the athlete that we hypothesize may have a positive impact on spastic hypertonia.

The aim of this case report is to cast light on go-kart practice effects on a patient with spastic SCI-related paraplegia, and to deepen understanding of which mechanisms could determine these effects by evaluating both the role of the vibratory energy transmitted by the vehicle and possible neuroendocrine system activation due to the drive.

## 2. Case Presentation

### 2.1. Patient’s Information

A 50-year-old male presented to Bari Maugeri Hospital in December 2022. He had suffered from spastic paraplegia since 2006 because of a traumatic SCI deriving from several vertebral fractures (from the sixth to the ninth dorsal vertebrae). He had a complete spinal cord lesion (ASIA A) and a functional independence measure (FIM) score of 75 (40 for the motor subscale and 35 for the cognition subscale). After the acute phase, which lasted for 6 months from the trauma, he developed a spasticity of the lower limbs. He had regularly undergone rehabilitation techniques, like stretching exercises and passive mobilization, for the first two years only. However, at the point of our evaluation, he was not undergoing any rehabilitative therapy and was not taking any anti-spastic drug.

For 15 years, the patient had regularly practiced go-kart racing and use of a dynamic driving simulator. The patient, after a complete go-kart driving session, reported significant transitory reductions in spastic hypertonia that lasted for the following 20 days. We sought to investigate and to objectively characterize the effect reported by the patient.

### 2.2. Materials and Methods

#### 2.2.1. Go-Kart Driving Session

With the aim of reproducing a normal driving session, the patient performed a two-hour drive session on the kart track (Figure 1).

The vehicle was equipped with two levers on the sides of the steering wheel that acted as accelerator and brake. These replaced the pedals commonly mounted on standard vehicle for able-body pilots which are not suitable for people with limitations (Figure 2).

Go-karts, because of their frame rigidity and engine characteristics, transmit vibration of poorly determined intensity, which may vary according to the engine features and displacement. Our patients’ vehicle was equipped with a 125 cubic centimeters displacement two-stroke engine. The patient performed the driving session at a maximum speed of 150 km per hour in a regulamentar kart track where no other drivers were admitted for the entire session.

#### 2.2.2. Spasticity Evaluation

The patient underwent a medical spasticity evaluation before (T0), immediately after (T1), 2 weeks after (T2), and 4 weeks after (T3) the go-kart session.

On both sides, long adductor, femoral bicep and medial and lateral gastrocnemius spasticity were assessed using the Modified Ashworth Scale (MAS). This is a widely used and validated scale for the clinical assessment of spasticity [28].

Each level of tone is given a score: 0 for no muscle tone increase; 1 for modest increase in muscle tone, with minimal resistance at the end of the range of motion in passive flexion or extension; 1+ for modest increase in muscle tone, with minimal resistance for less than half of the range of motion; 2 for marked increase in muscle tone during most of the joint range of motion, with passive movements still possible; 3 for significant increase in muscle tone, with difficult passive movement; 4: for limb rigidity in both flexion and extension [29].

To perform the measurements, with the patient in the supine position, the examiner tested the long adductors of both legs, moving the hips from the maximal adduction position to a maximal abduction one; with the patient in the prone position, the examiner evaluated the femoral biceps on both sides, moving the knee joint from a maximal flexed position to a maximal extended one; finally, to assess the MAS score for the lateral and medial gastrocnemius, the examiner moved the ankle joint from a maximally extended position to a position of maximal flexion over one second.

Then, the examiner bilaterally assessed the tone and stiffness of the femoral biceps, the long adductors, the medial gastrocnemius, and the lateral gastrocnemius using MyotonPro.

MyotonPRO is a non-invasive device used to measure tone and dynamic stiffness [30]. The oscillation frequency (measured in Hertz) represents the intrinsic tension of the biological soft tissues, in particular, the tone of the superficial skeletal muscles in their passive or resting state with no voluntary contraction; by contrast, the dynamic stiffness (measured in Newtons/meter) is the resistance of biological tissues to a deformation force [31].

The MyotonPRO device is applied under a stable preload (0.18 Newtons) to superficial tissues. It is placed perpendicular to the muscle to be investigated, and involves application of a short (15 ms) mechanical tap at a preset force (0.4 Newtons), followed by quick release, producing oscillations that are collected by the testing probe. The muscle vibrations are then recorded using an accelerometer, and dedicated software translates these vibrations into numerical values, which represent muscle tone and biomechanical characteristics. The accuracy of this tool has been confirmed in the available scientific literature [32,33].

These measurements were undertaken with the patient in a supine position, with the hips in a neutral position and the knees not fully extended because of spasticity. The test landmarks for the long adductors were at two-thirds of the distance between the anterior superior iliac spine and the middle third of the sour femoral line.

For the femoral biceps, with the patient in a prone position, the landmarks were at two-thirds of the distance between the ischial tuberosity and the head of the fibula; the measurements for the lateral and medial gastrocnemius were taken with the patient in a prone position at one-third of the distance between the popliteal fossa and the Achilles tendon, one in the medial part and the other in the lateral part of the muscle. To increase the reliability of the measurement, the landmarks were labeled with a marker harmless to the human body.

All these measurements were performed by a physician specialized in physical and rehabilitation medicine with twenty years experience in the treatment of spasticity (Figure 3 and Figure 4).

## 3. Results

The examiner reported that, at T0, the long adductors and the femoral biceps had a MAS score of 3 and the gastrocnemius had a score of 2 bilaterally; at T1, a significant reduction in spasticity was recorded: the MAS score was 1+ in all the examined muscles, except for the left gastrocnemius whose score was 1. At T2, all these values were slightly raised, and at T3 they returned to be superimposable on those of T0 (Table 1).

The same trend was recorded for the muscle tone and stiffness scores: at T1, all the muscle tone results were considerably lower than at T0, then there was a modest increase at T2 and a definite rise at t3, when the values were similar to those recorded at T0 (Table 2);

The stiffness values at T1 were markedly less than at T0; then they showed a small increase at T2 and a final raise was observed at T3, with values similar to those at T0 (Table 3).

## 4. Discussion

The present case report is interesting since it shows that a go-kart drive session could reduce muscle spasticity in a paraplegic patient who had an SCI. In particular, we found a spasticity improvement according to the MAS and a reduction in terms of muscle tone and stiffness according to MyotonPRO measurements. All these changes gradually regressed until a substantial return was observed to the initial values at T3.

But, what was the mechanism underlying these transitory muscle changes?

We hypothesized two possible mechanisms.

### 4.1. WBV Hypothesis

Firstly, go-kart driving exposed the patient to a quantity of intense vibrations comparable to a WBV session.

Go-karting is a motorsport specialty which involves racing with karts. These are small compact vehicles with minimum ground clearance and without suspension, a flexible chassis, rear-wheel drive and an engine with limited power, which, however, make it possible to reach speeds of 100 km/h. These vehicles deliver an intense, non-measurable, and long-lasting dose of WBV, which is transmitted to the athlete’s body; this could reproduce the positive effect of WBV on spastic hypertonia.

Although there is still little evidence to support the benefits of WBV on spasticity in people with SCI and the appropriate protocols of WBV are still debated, this hypothesis requires further analysis. In fact, our results are in line with some recent scientific findings [34,35]. It seems that vibrations elicit a motor neuron discharge increment and H-reflex depression [36,37]. These stimuli excite spinal circuits by activating muscle spindles and alpha-motor neurons, thus synchronizing motor units [38]. As a consequence, with rapid and repetitive movements, the vibrations improve muscle voluntary activation, reducing spasticity because of the alteration of afferent neuron discharges and increase in motor unit recruitment [39]. Furthermore, go-kart driving, like WVB, can also have vasodilatory effects that may lead to an increase in cellular metabolism with consequent increase in muscle activity, also reducing spasticity [40,41]. Ness et al. performed a 4-week, 12-session program of WBV therapy with a group of SCI patients with spastic hypertonia of the lower limbs; they found a reduction in quadriceps spasticity and a positive effect on walking speed, cadence, and step length [42]. WBV has also been observed to be effective in improving strength and mobility and in reducing pain and stiffness in neurologic diseases other than SCI, such as stroke, multiple sclerosis, and cerebral palsy [43,44,45]. Sayenko et al. showed that a single 1 min WBV session could inhibit the soleus H-reflex just by passive standing on a WBV plate; this suggests that acute modulation of spinal motoneuronal excitability can be achieved without voluntary leg muscle contraction, with positive implications for individuals with neurological impairments and compromised muscular contractions, such as SCI patients [46].

With regard to the WBV mechanism on spasticity, it has also been suggested that the H-reflex might be suppressed due to muscle fatigue caused by this therapy since there is physiological evidence that the normally fatigue-resistant soleus fibers transform to faster fatigable muscle as a result of fiber-type transformation after SCI [47].

Our results are also in line with the available literature in terms of the duration of the muscle relaxant effect. In fact, a systematic review by Sadeghi et al. demonstrated that WBV reduced spasticity for a period of 6–8 days after a vibration session [48].

Driving a go-kart would make it possible to achieve results similar to WBV in spasticity reduction while still carrying out a sporting activity which is fun, enabling patients to obtain general psychophysical benefits.

### 4.2. Neuroendocrine Hypothesis

Our second hypothesis to account for the abovementioned results concerns the neuroendocrine reaction that karting can induce as a sport.

It is well known that physical exercise and sport can increase blood levels of monoamines, neurotropins, and neuromodulators, such as endogenous opioids and endocannabinoids [49,50,51,52].

Could these molecules transitorily reduce spasticity after sport practice?

Regarding endocannabinoids, they regulate synaptic neurotransmission and can control neurological symptoms like spasticity and tremor [53]. These effects have been investigated particularly in multiple sclerosis patients [54,55], but they may be beneficial in a wide range of neurological diseases due to their neuromodulatory effect on spastic muscles [56]. Similarly, serotonin [57] and other monoamines [58] are produced in response to sporting activities and have a muscle relaxant action that could also counteract spasticity [59].

### 4.3. Limitations and Ideas for the Future

This study has many limitations. First of all, it is just a case report study. As a consequence, there is no control group. In order to deepen understanding of the relationship between go-kart driving and the documented changes in muscle spasticity, comparison with a group of SCI patients practicing different forms of physical activity, or with a group not engaging in any physical activity, would be needed, while increasing the sample size. We have already started this research, which will require more time due to the need to address ethical aspects related to the potential risks to which patients are exposed while driving. Similarly, a long-term follow up is needed to determine the duration over time of the muscle effects of driving a go-kart, even if these effects appear transient. Moreover, to better investigate our neuroendocrine hypothesis, further studies based on investigating blood dosages of hormones are necessary.

On the other hand, we consider a strength of this study to be the fact that, to our knowledge, for the first time, go-kart driving has been considered as a potential therapeutic option for spasticity treatment.

## 5. Conclusions

Go-kart driving may represent a new opportunity for the rehabilitation of patients with motor disabilities resulting from SCI or other neurological pathologies, even though the mechanisms underlying the transient reduction in muscle spasticity are not clear. We hope that our experience can provide a starting point for further studies of higher quality to confirm our results and to verify whether they can be extended to larger samples.

## Figures and Tables

**Figure 1 diseases-11-00107-f001:**
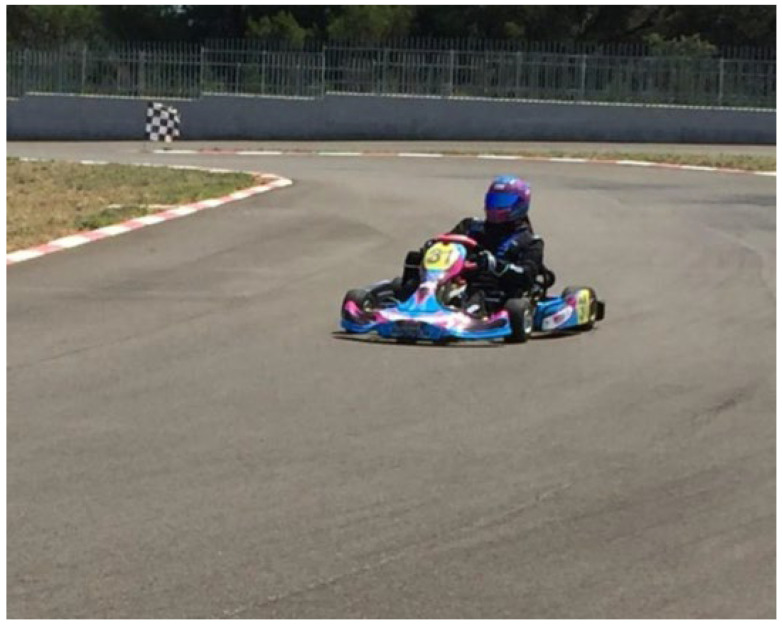
A go-kart driving session on the track.

**Figure 2 diseases-11-00107-f002:**
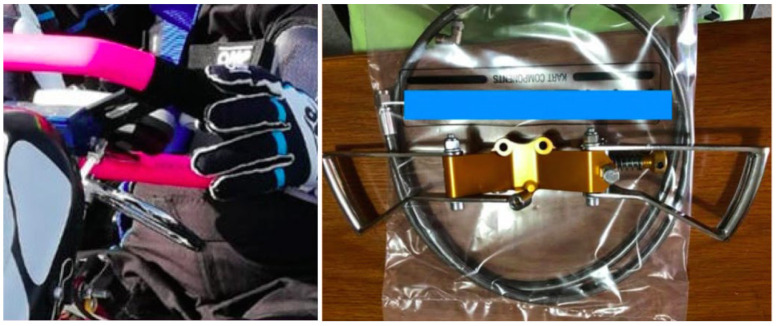
Controls adapted to the steering for SCI drivers.

**Figure 3 diseases-11-00107-f003:**
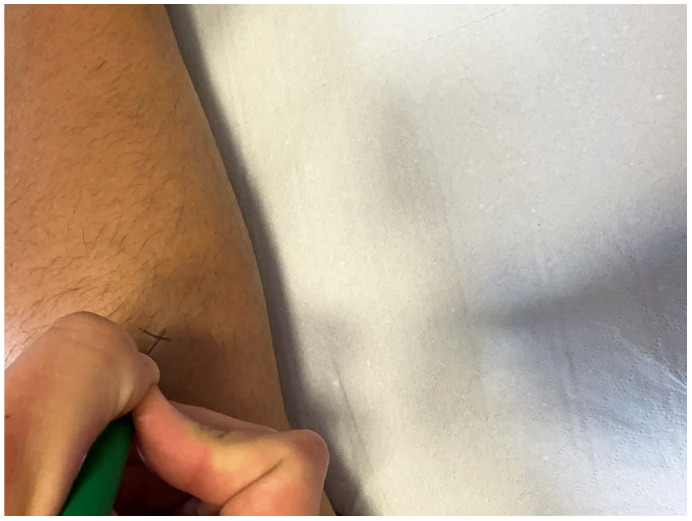
Left medial gastrocnemius landmark labeled with a harmless marker.

**Figure 4 diseases-11-00107-f004:**
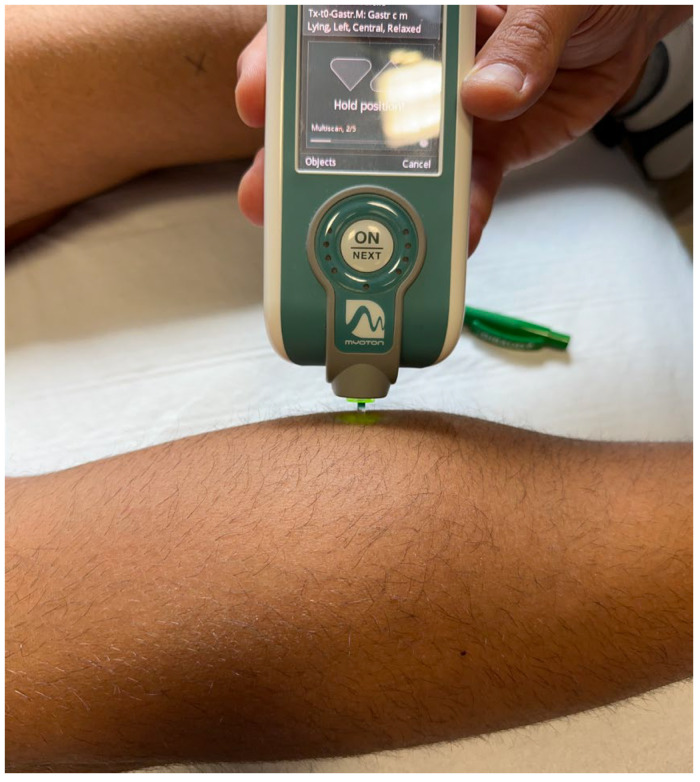
Left medial gastrocnemius tone and stiffness evaluation using MyotonPro.

**Table 1 diseases-11-00107-t001:** Muscle Modified Ashworth Scale (MAS) scores at all detection times.

Muscle	T0	T1	T2	T3
Right long adductor	3	1+	2	3
Left long adductor	3	1+	1+	3
Right femoral biceps	3	1+	1+	3
Left femoral biceps	3	1+	1+	2
Right gastrocnemius	2	1+	2	2
Left gastrocnemius	2	1	1+	2

**Table 2 diseases-11-00107-t002:** Muscle tone values measured in Hertz using MyotonPro, at all the detection times.

Muscle	T0	T1	T2	T3
Right long adductor	16.8 Hz	10.1 Hz	14.5 Hz	16 Hz
Left long adductor	17.2 Hz	8.6 Hz	13.8 Hz	16.8 Hz
Right femoral biceps	16.8 Hz	9.8 Hz	13 Hz	15.1 Hz
Left femoral biceps	11.7 Hz	7.3 Hz	10.3 Hz	11.2 Hz
Right medial gastrocnemius	10.7 Hz	6.5 Hz	8.8 Hz	10 Hz
Left medial gastrocnemius	10 Hz	6.1 Hz	8.5 Hz	9.7 Hz
Right lateral gastrocnemius	24.6 Hz	14 Hz	20.4 Hz	23 Hz
Left lateral gastrocnemius	25.1 Hz	15 Hz	20.8 Hz	24.2 Hz

**Table 3 diseases-11-00107-t003:** Muscle stiffness values measured as Newtons/meter using MyotonPro at all the detection times.

Muscle	T0	T1	T2	T3
Right long adductor	403 N/m	226 N/m	300 N/m	382 N/m
Left long adductor	347 N/m	205 N/m	268 N/m	293 N/m
Right femoral biceps	223 N/m	160 N/m	181 N/m	199 N/m
Left femoral biceps	342 N/m	202 N/m	256 N/m	322 N/m
Right medial gastrocnemius	317 N/m	273 N/m	288 N/m	300 N/m
Left medial gastrocnemius	344 N/m	304 N/m	392 N/m	333 N/m
Right lateral gastrocnemius	397 N/m	300 N/m	362 N/m	378 N/m
Left lateral gastrocnemius	395 N/m	286 N/m	300 N/m	384 N/m

## Data Availability

Not applicable.

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
