# Peer review of "Is There a New Road to Spinal Cord Injury Rehabilitation? A Case Report about the Effects of Driving a Go-Kart on Muscle Spasticity"

_diseases, 2023, doi:10.3390/diseases11030107_

Round 1
Reviewer 1 Report
Title: Is there a new road to spinal cord injury rehabilitation? A case report about the effects of driving Go-Kart on muscle spasticity
Abstract: The paper presents a case study investigating the effects of Go-Kart driving on muscle spasticity in a patient with spinal cord injury (SCI). The patient, a 50-year-old male with spastic paraplegia due to traumatic SCI, underwent Go-Kart driving sessions and was evaluated at various time points. The results suggest that Go-Kart driving sessions could transiently reduce muscle spasticity, tone, and stiffness. The authors hypothesize two possible mechanisms underlying these changes: intense whole-body vibration (WBV) similar to that experienced during Go-Kart driving and neuroendocrine reactions induced by physical exercise. The paper concludes that Go-Kart driving might offer a novel approach to SCI rehabilitation, and further research is warranted.
General Assessment: The paper presents an interesting case study exploring the potential effects of Go-Kart driving on muscle spasticity in an SCI patient. The topic is relevant, considering the growing interest in adapted sports for individuals with disabilities. The study design is appropriate for a case report, and the data collection methods seem adequate. However, the paper would benefit from some improvements in several areas.
Major Comments:
-
Lack of Control Group: The major limitation of this study is the absence of a control group. Without a control group, it is challenging to establish a cause-and-effect relationship between Go-Kart driving and the observed changes in muscle spasticity. A comparison with a group of SCI patients engaging in a different form of physical activity or a group not engaging in any physical activity would strengthen the findings and allow for more robust conclusions.
-
Sample Size and Generalization: The study is limited to a single patient, which severely restricts the generalizability of the results. One patient cannot represent the diverse population of individuals with SCI. Future studies should aim for a larger sample size, ideally with participants from different age groups, injury levels, and spasticity severities to assess the true impact of Go-Kart driving on muscle spasticity in a broader context.
-
Lack of Long-Term Follow-up: The study evaluated the patient's muscle spasticity at specific time points after Go-Kart driving sessions, but the long-term effects were not investigated. A more extended follow-up period would be valuable to determine the sustainability and persistence of the observed effects.
-
Mechanism Exploration: The paper provides two plausible hypotheses for the observed changes: intense WBV and neuroendocrine reactions. However, these hypotheses are not directly tested or explored further. Including additional experiments or measures related to these mechanisms could provide more substantial evidence for the proposed explanations.
Minor Comments:
-
Clarity and Organization: The paper could be improved by enhancing the clarity of presentation and organization. Some sentences and paragraphs are overly complex and require better structure. The authors should ensure that the information flows logically and is easy for readers to follow.
-
Citations and References: The authors make references to previous scientific findings but do not provide proper citations or a complete reference list. This oversight should be addressed to allow readers to access the relevant literature and verify the claims.
Moderate editing of English language required
Author Response
Dear Reviewer,
Thank you for your eminent and valuable revision. You gave us a global and interesting scientific vision. We really appreciated your comments, which certainly will help us to improve the quality of our article.
Here are our point to point replies.
Major Comments:
- R: Lack of Control Group: The major limitation of this study is the absence of a control group. Without a control group, it is challenging to establish a cause-and-effect relationship between Go-Kart driving and the observed changes in muscle spasticity. A comparison with a group of SCI patients engaging in a different form of physical activity or a group not engaging in any physical activity would strengthen the findings and allow for more robust conclusions.
A: thank you for this suggestion. We integrated the last section of the article clearly explaining the limitations of this study. As we specified, this was just a case report study. As a consequence, there is no control group, because we enrolled only a patient. In order to deepen the relationship between Go-Kart driving and the documented changes in muscle spasticity, a comparison with a group of SCI patients practicing different form of physical activity or with a group not engaging in any physical activity would be needed, and we are already working in this sense. As you know, this kind of research requires more time due to the ethical aspects related to the potential risks to which fragile patients are exposed while driving.
- R: Sample Size and Generalization: The study is limited to a single patient, which severely restricts the generalizability of the results. One patient cannot represent the diverse population of individuals with SCI. Future studies should aim for a larger sample size, ideally with participants from different age groups, injury levels, and spasticity severities to assess the true impact of Go-Kart driving on muscle spasticity in a broader context.
A: For the same reasons mentioned above, the sample size should be enlarged in further studies designed as more complex and structured case-control trials.
- R: Lack of Long-Term Follow-up: The study evaluated the patient's muscle spasticity at specific time points after Go-Kart driving sessions, but the long-term effects were not investigated. A more extended follow-up period would be valuable to determine the sustainability and persistence of the observed effects.
A: we agree that a long term follow up is needed to verify the duration over time of the muscles effects of the Go-kart driving. Nevertheless, according to our findings these effects appear transient. A larger sample and a longer follow up could anyway deepen the duration of the effects.
- R: Mechanism Exploration: The paper provides two plausible hypotheses for the observed changes: intense WBV and neuroendocrine reactions. However, these hypotheses are not directly tested or explored further. Including additional experiments or measures related to these mechanisms could provide more substantial evidence for the proposed explanations.
A: we really appreciated this comment. To better investigate our neuroendocrine hypothesis, further studies based on blood dosages of hormones are necessary. As we said, this was just a case report, a starting point for our further researches which will be more structured.
Minor Comments:
- R: Clarity and Organization: The paper could be improved by enhancing the clarity of presentation and organization. Some sentences and paragraphs are overly complex and require better structure. The authors should ensure that the information flows logically and is easy for readers to follow.
A: Thank you for this suggestions. We revised globally the structure of the paper to make it easier to read and understand.
- R: Citations and References: The authors make references to previous scientific findings but do not provide proper citations or a complete reference list. This oversight should be addressed to allow readers to access the relevant literature and verify the claims.
A: We revised also the bibliography, which was already at the end of the manuscript file.
We also revised English language according to your suggestion.
Thank you in advance for the time you dedicated to our paper and for the effort to guide us in improving it.
Best regards
Reviewer 2 Report
This manuscript was focusing on effects of Go-Kart on the patient with SCI-related spasticity, the topic looks interesting. My comments are as follows:
1) Both motivations and contributions are unclear in Abstract and Introduction, please refine them.
2) Separate related work section to review state-of-the-arts in this area should be considered by the authors
3) High-quality figures are strongly suggested to better demonstrate the proposed method and experimental results.
4) The experiments are not sufficient. More baseline methods, datasets, scenarios and evaluation metrics should be included to support the proposed method.
5) The authors should comprehensively discuss both the limitations of the proposed method and future directions. For example, how to use deep learning to assist the rehabilitation? Some related papers are recommended, which are better included in the reference list: physics-informed deep learning for musculoskeletal modeling: Predicting muscle forces and joint kinematics from surface emg, IEEE TNSRE, and non-iterative and fast deep learning: multilayer extreme learning machines, JFI.
English writing improvement should be considered by the authors.
Author Response
Dear Reviewer,
Thank you for your eminent and valuable revision. You gave us a global and interesting scientific vision. We really appreciated your comments, which certainly will help us to improve the quality of our article.
Here are our point to point replies.
- R: Both motivations and contributions are unclear in Abstract and Introduction, please refine them.
A: Thank you for this suggestion. We integrated and clarified what is the purpose of this manuscript. In particular we underlined that our aim is to investigate the benefits of Go-Kart driving as an adapted sport for patients with SCI-related spasticity, deepening both the mechanism of WBVs and the neuroendocrine hypothesis.
- R: Separate related work section to review state-of-the-arts in this area should be considered by the authors
A: We proceeded to separate the working sections in the discussion in order to highlight what is the state of the art. In this way we highlighted the WBV mechanism, the hormonal one and the ideas for further studies.
- R: High-quality figures are strongly suggested to better demonstrate the proposed method and experimental results.
A: thank you for this suggestion, we added three figures to better explain the proposed method
- R: The experiments are not sufficient. More baseline methods, datasets, scenarios and evaluation metrics should be included to support the proposed method.
A: thank you for this suggestion. We integrated the last section of the article clearly explaining the limitations of this study. As we specified, this was just a case report study. We integrated some information about our patient, Go-Kart, truck and driving session. We added some figures to better explain the proposed methods.
- R: The authors should comprehensively discuss both the limitations of the proposed method and future directions. For example, how to use deep learning to assist the rehabilitation? Some related papers are recommended, which are better included in the reference list: physics-informed deep learning for musculoskeletal modeling: Predicting muscle forces and joint kinematics from surface emg, IEEE TNSRE, and non-iterative and fast deep learning: multilayer extreme learning machines, JFI.
A: Regarding the limitations and future direction, we believe that a long-term follow-up and a larger patients sample are needed to verify the duration of the muscular effects of Go-kart driving over time. We highlighted this aspects in a dedicated paragraph, as said above. We also thank you for the suggested studies, we considered them useful to improve the introduction of this paper.
We also revised the English language with the help of an English mother tongue Academic Professor.
Thank you in advance for the time you dedicated to our paper and for the effort to guide us in improving it.
Best regards
Reviewer 3 Report
Authors present an interesting case report about a possible innovative rehabilitation treatment for traumatic spinal cord injury: Go-kart. On the basis of data collected Go- Kart driving session could reduce muscle spasticity, tone and stiffness.
The paper is well written and very interesting. I just suggest to cite the importance of premorbid physical activity and the role of muscle strenght (10.1016/j.archger.2020.104109) in traumatic event.
Thanks
Author Response
Dear Reviewer,
Thank you for your eminent and valuable revision. You gave us a global and interesting scientific vision. We
really appreciated your comments, which certainly will help us to improve the quality of our article.
In particular, thank you for the interesting advice to highlight the importance of premorbid physical activity
and the role of muscle strenght, we considered the article you suggested a great occasion to integrate our
paper.
Thank you in advance for the time you dedicated to our paper and for the effort to guide us in improving it.
Best regards
Reviewer 4 Report
Is there a new road to spinal cord injury rehabilitation? A case 2 report about the effects of driving Go-Kart on muscle spasticity
Abstract
· Add that this is a case report
Introduction
· Add information on the effectiveness of commonly used spasticity interventions
· Add more explanations on what is Go Cart and its history in SCI
· Add more on Go Cart studies in SCI in general and more specifically on spasticity.
· Why you think that go cart will have an influences on spasticity? What is the scientific rational?
Methods
· Can you provide more information on the patient using validated scales? ASIA level? Spasticity tests? Functional level? And more?
· What did the medical evaluations consisted of?
· Can you have a separate title for the evaluations conducted and separate one for the description of the activity conducted
· Describe better what a driving session looks like
· Because it is one patient in order to say that there was an increase or decrease it will be helpful adding info on clinical meaningful change in the various outcomes. If they exist.
Results
· How many sessions did he have? How long were the sessions?
· Add measurement units to tables
Discussion
· I didn’t see T3 in the table
· Add more references where it is missing. To have one at least in each paragraph.
Conclusion
Net of the mechanism underlying the transient reduction of muscle spasticity – it is not clear to me.
Author Response
Dear Reviewer,
Thank you for your eminent and valuable revision. You gave us a global and interesting scientific vision. We really appreciated your comments, which certainly will help us to improve the quality of our article.
Here are our point to point replies.
Abstract
R: Add that this is a case report
A: Thank you for this suggestion. We added the information in the background section.
Introduction
R: Add information on the effectiveness of commonly used spasticity interventions
A: Thank you for this suggestion. We integrated the introduction section according to it.
R: Add more explanations on what is Go Cart and its history in SCI
A: sorry for this but no history of Go Kart in SCI is available in literature.
R: Add more on Go Cart studies in SCI in general and more specifically on spasticity.
A: there is no scientific literature available on this topic, the case report represent a starting point for further studies.
R: Why you think that go cart will have an influences on spasticity? What is the scientific rational?
A: Thank you for this suggestion, we tried to better define the scientific rational in the introduction and to better explain it in the discussion
Methods
R:Can you provide more information on the patient using validated scales? ASIA level? Spasticity tests? Functional level? And more?
A: We used MyotonPro and MAS, a part from a general medical evaluation. Moreover,the patient had a complete lesion (ASIA A) and a FIM scale total score of 75 ( 40 for motor subscale and 35 for cognition subscale)
R: What did the medical evaluations consisted of?
A:Before the spasticity evaluation using MAS scale , we collected a short medical history and evaluated patiets’ ROM, strenght by using RMS, patellar reflex, achilles reflex and plantar reflex and sensibility both deep and superficial.
R: Can you have a separate title for the evaluations conducted and separate one for the description of the activity conducted
A: Thank you for this suggestions. We revised the structure of the matherials and methods to make it easier to read and understand.
R: Describe better what a driving session looks like
A: We better describe it, as you required.
R: Because it is one patient in order to say that there was an increase or decrease it will be helpful adding info on clinical meaningful change in the various outcomes. If they exist.
A: no other clinical changes were reported.
Results
R: How many sessions did he have? How long were the sessions?
A:He had only one long session lasting two hours
R: Add measurement units to tables
A: Thank you for this suggestion, we added the measurement units on tables
Discussion
R: I didn’t see T3 in the table
A: Sorry for this, T3 was in the table, we fixed the layout problem in the MDPI document.
R: Add more references where it is missing. To have one at least in each paragraph.
A: We revised also the bibliography, which was already at the end of the manuscript file.
Conclusion
R: Net of the mechanism underlying the transient reduction of muscle spasticity – it is not clear to me.
A: Thank you for the suggestion, we modified the conclusion to make easier to understand our conclusion” Go-Kart driving may represent a new opportunity for the rehabilitation of patients with motor disabilities resulting from SCI or other neurological pathologies, despite the mechanisms underlying the transient reduction of muscle spasticity are not clear”.
Thank you in advance for the time you dedicated to our paper and for the effort to guide us in improving it.
Best regards
Round 2
Reviewer 1 Report
Dear Authors,
I have thoroughly reviewed your manuscript and I'm delighted to inform you that you have effectively addressed all of my queries and suggestions. The revisions made to the manuscript are commendable and have greatly improved its quality. Considering the substantial improvements you've implemented, I am pleased to highly recommend accepting your paper for publication in the Journal of Diseases. Your dedication to enhancing the manuscript is truly commendable. Keep up the excellent work!
Best regards,
Reviewer 2 Report
The authors have addressed the issues, I have no more comments.
No comments.